# A diverse host thrombospondin-type-1 repeat protein repertoire promotes symbiont colonization during establishment of cnidarian-dinoflagellate symbiosis

**Emilie-Fleur Neubauer[1], Angela Z Poole[2,3], Philipp Neubauer[4], Olivier Detournay[5], Kenneth Tan[3], Simon K Davy[1]\*, Virginia M Weis[3]\***

[1]School of Biological Sciences, Victoria University of Wellington, Wellington, New Zealand; [2]Department of Biology, Western Oregon University, Monmouth, United States; [3]Department of Integrative Biology, Oregon State University, Corvallis, United States; [4]Dragonfly Data Science, Wellington, New Zealand; [5]Planktovie sas, Allauch, France

**Abstract** The mutualistic endosymbiosis between cnidarians and dinoflagellates is mediated by complex inter-partner signaling events, where the host cnidarian innate immune system plays a crucial role in recognition and regulation of symbionts. To date, little is known about the diversity of thrombospondin-type-1 repeat (TSR) domain proteins in basal metazoans or their potential role in regulation of cnidarian-dinoflagellate mutualisms. We reveal a large and diverse repertoire of TSR proteins in seven anthozoan species, and show that in the model sea anemone *Aiptasia pallida* the TSR domain promotes colonization of the host by the symbiotic dinoflagellate *Symbiodinium minutum*. Blocking TSR domains led to decreased colonization success, while adding exogenous TSRs resulted in a 'super colonization'. Furthermore, gene expression of TSR proteins was highest at early time-points during symbiosis establishment. Our work characterizes the diversity of cnidarian TSR proteins and provides evidence that these proteins play an important role in the establishment of cnidarian-dinoflagellate symbiosis.

**\*For correspondence:** Simon. Davy@vuw.ac.nz (SKD); weisv@ oregonstate.edu (VMW)

**Competing interests:** The authors declare that no competing interests exist.

## Introduction

Host-microbe interactions, both beneficial and detrimental, are ancient and ubiquitous, and are mediated by a myriad of molecular and cellular signalling events between the partners. Hosts are under selective pressures to develop recognition mechanisms that tolerate beneficial symbionts and destroy negative invaders, while microbes evolve to successfully invade and either benefit or exploit their hosts (*Eberl, 2010*; *Bosch and McFall-Ngai, 2011*). Cnidarian-dinoflagellate mutualisms, such as those that form coral reefs, are one such host-microbe interaction for which we are just beginning to uncover the molecular conversations between partners that result in the establishment and maintenance of a healthy partnership (*Davy et al., 2012*). Most cnidarian-dinoflagellate partnerships are established anew with each cnidarian host generation. The photosynthetic dinoflagellates (*Symbiodinium* spp.) are taken from the environment into host gastrodermal cells *via* phagocytosis and, instead of being digested, the symbionts persist and colonize the host.

Discovery-based, high-throughput 'omics' techniques have previously been employed to uncover candidate genes and pathways that could play a role in inter-partner recognition and regulation

**eLife digest** Cnidarians, such as corals and sea anemones, often form a close relationship with microscopic algae that live inside their cells – a partnership, on which the entire coral reef ecosystem depends. These microalgae produce sugars and other compounds that the cnidarians need to survive, while the cnidarians protect the microalgae from the environment and provide the raw materials they need to harness energy from sunlight. However, very little is known about how the two partners are able to communicate with each other to form this close relationship, which is referred to as a symbiosis.

Symbiotic relationships between a host and a microbe require a number of adaptations on both sides, and involve numerous signalling molecules. A host species is under constant pressure to develop mechanisms to recognize and tolerate the beneficial microbes without leaving itself vulnerable to attack by microbes that might cause disease. Similarly, the beneficial microbes need to be able to invade and survive inside their host.

Previous research has shown that TSR proteins in hosts play a role in recognizing and controlling disease-causing microbes. Until now, however, it was unknown whether TSR proteins are involved in establishing a symbiosis between cnidarians and their algal partners.

Neubauer et al. analysed six species of symbiotic cnidarians and discovered a diverse repertoire of TSR proteins. These proteins were found in the host genomes, rather than in the symbiotic algae, strongly suggesting that they originated from the host.

Neubauer et al. next incubated a sea anemone species in a solution of TSR proteins and saw that it became 'super-colonized' with algae, meaning that over time, millions of the microalgae entered and stayed in the anemone's tentacles. In contrast, when the TSR proteins were blocked, colonization was almost entirely stopped. This suggests that host TSR proteins play an important role for the microalgae when they colonialize corals and other cnidarians.

The signals that enable microalgae to successfully colonize cnidarians are unquestionably complex and there is still much to learn. These findings add another piece to the puzzle of how symbiotic algae bypass the cnidarian's immune system to persist and flourish in their host. An important next step will be to test how blocking the genes that encode the TSR proteins will affect the symbiotic relationship between these species.

processes in cnidarian-dinoflagellate symbioses (*Meyer and Weis, 2012*; *Mohamed et al., 2016*). Two such transcriptomic studies comparing expression patterns of symbiotic and aposymbiotic individuals of the sea anemone species *Anthopleura elegantissima* and *Aiptasia pallida* (*Rodriguez-Lanetty et al., 2006*; *Lehnert et al., 2014*), started us down a path to an in-depth examination of thrombospondin-type-1-repeat (TSR)-domain-containing proteins (hereafter referred to as TSR proteins) in both partners of the symbiosis. Both studies found significant upregulation of a homologue to a scavenger receptor type B1 (SRB1) in symbiotic anemones. The structure and diversity of SRB1s have now been characterized in a variety of cnidarians, including *A. elegantissima* and *A. pallida* (*Neubauer et al., 2016*). SRB1s function in innate immunity in metazoans in a variety of ways, including, in mammals, activation of the tolerogenic, immunosuppressive transforming growth factor beta (TGF$\beta$) pathway (*Asch et al., 1992*; *Masli et al., 2006*; *Yang et al., 2007*). When the TSR domains of the extracellular matrix glycoprotein thrombospondin bind to CD36, latent TGF$\beta$ is converted to its active form, which in turn launches tolerogenic pathways downstream. Subsequent studies in another sea anemone model system, *A. pallida*, demonstrated a role for TGF$\beta$ in the regulation of cnidarian-dinoflagellate symbioses (*Detournay et al., 2012*). This warranted further examination of genes related to TGF$\beta$ pathway activation and turned our focus to thrombospondins.

Our initial search for thrombospondin and other TSR protein homologues revealed a rich literature on thrombospondin-related anonymous proteins (TRAPs) that play important roles in apicomplexan endoparasites, such as when *Plasmodium* attaches to and invades mammalian host cells (*Kappe et al., 1999*; *Vaughan et al., 2008*; *Morahan et al., 2009*). Specifically, the WSPCSVTCG motif (*Figure 1*) within the TRAP TSR binds sulfated glycoconjugates on host cells (*Morahan et al., 2009*). This piqued our interest in TSRs even more, because apicomplexans and dinoflagellates are

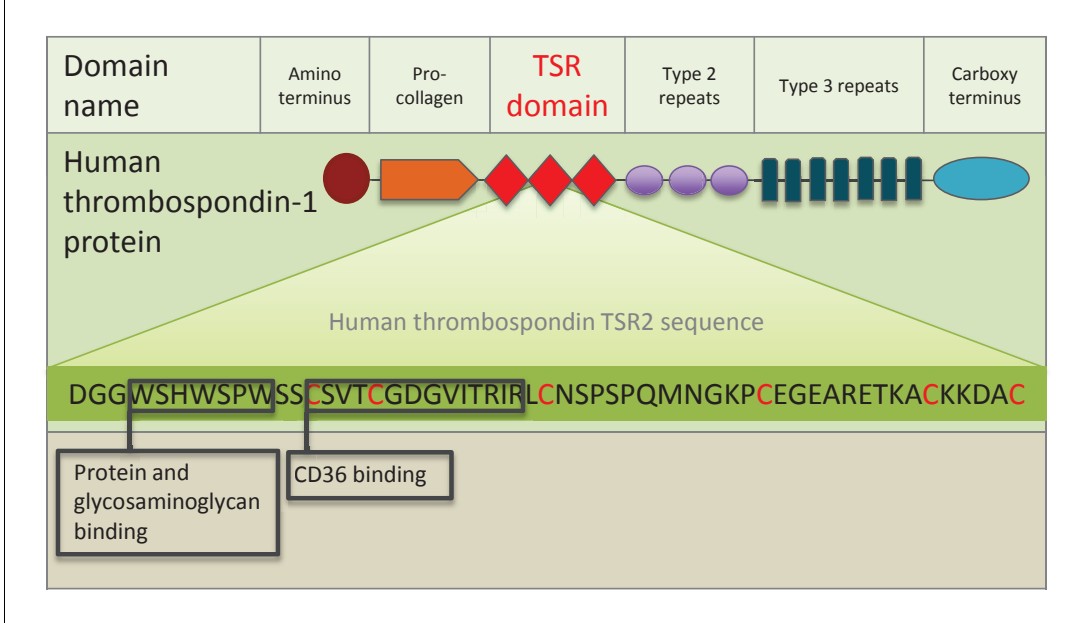

**Figure 1.** Schematic representation of human thrombospondin 1 protein. The three TSR (Thrombospondin Structural homology Repeat) domains are depicted by three red diamonds. The amino acid sequence of the second TSR sequence is shown with six conserved cysteines in red. Known binding motifs and capabilities of the human thrombospondin TSR domain two are listed and depicted in boxes. (Redrawn from *Zhang and Lawler, 2007*).

sister taxa within the alveolates (*Burki et al., 2008*; *Adl et al., 2012*). There is therefore the potential for homologous strategies of symbiont invasion and persistence in hosts that spurred our interest in a deeper investigation of TSR homologues within *Symbiodinium*, as well as within host cnidarians.

The large TSR protein superfamily includes mammalian thrombospondins (depicted in *Figure 1*), and many proteins in metazoans and other eukaryotes (*Adams and Tucker, 2000*; *Tucker, 2004*). The superfamily is composed of secreted and transmembrane proteins with a large array of functions involving protein-protein and other steric interactions. TSR superfamily members are diverse, suggesting that the highly-conserved TSR domain has been duplicated and shuffled numerous times among superfamily members. For example, 41 human genes contain one or more TSR domain copies (*Silverstein, 2002*), while there are 27 and 14 TSR superfamily members in *C. elegans* and *Drosophila*, respectively (*Tan et al., 2002*). The TSR domain consists of approximately 60 amino acids (*Figure 1*), with several highly conserved motifs and five or six conserved cysteine residues that participate in disulfide bridge formation and domain folding (*Adams and Lawler, 2011*).

Thrombospondins were originally characterized in mammals. They are extracellular, multi-domain, calcium-binding glycoproteins that play pleiotropic tissue-specific roles involving interactions with cell surfaces, cytokines and the extracellular matrix (*Adams and Lawler, 2004*). Protein-protein interactions involving the TSR domain, including binding to SRB1/CD 36 (see *Figure 1*), are central to thrombospondin protein function. A systematic search for TSR proteins across the Cnidaria has not been conducted to date. However, a study of vertebrate thrombospondin protein homologues in *Nematostella vectensis* found that, although most of the multi-domain architecture is present, crucially, the three TSR domains are missing (thus adding a confusing naming problem to the categorization of these genes) (*Bentley and Adams, 2010*; *Tucker et al., 2013*).

There is, however, growing evidence that cnidarians possess numerous genes that contain TSR domains. Two rhamnospondin genes with eight TSR domains were identified in the colonial hydroid *Hydractinia symbiolongicarpus* that are expressed in the hypostome of feeding polyps and were proposed to function in microbe binding (*Schwarz et al., 2007*). A study in *Hydra oligactis* also demonstrated high expression of several genes for TSR proteins in the hypostome and proposed potential functions in nerve net development or defense (*Hamaguchi-Hamada et al., 2016*). Within anthozoans, several TSR proteins were identified in two species of corals, *Acropora palmata* and

*Montastraea faveolata* (*Schwarz et al., 2008*), and in a study identifying candidate symbiosis genes across ten cnidarian species (*Meyer and Weis, 2012*). Therefore, while a number of studies have focused on characterization and localization of cnidarian TSR proteins, their proposed functions have not yet been investigated.

The aim of this study was to characterize and compare the TSR protein repertoire of seven cnidarian species (six symbiotic, one non-symbiotic) and two symbiotic dinoflagellate species, to identify putative ligands for SRB1/CD36 in host sequence resources and TRAP-like proteins in the *Symbiodinium* genome. Using six anthozoan genomic and transcriptomic resources, we compared vertebrate TSR proteins of known function with the cnidarian TSR repertoire. We investigated the presence of known binding motifs and their conservation within the cnidarian TSR domains. In addition, we explored the function of TSR proteins in cnidarian-dinoflagellate symbiosis, using the sea anemone *A. pallida*, a globally-adopted model system for the study of this symbiosis (*Weis et al., 2008*; *Goldstein and King, 2016*). We tested the hypothesis that TSR proteins are involved in symbiont colonization of the host during onset of symbiosis, and whether the proteins of interest are of host or symbiont origin. Functional studies were performed in which TSR-domain function was blocked, or exogenous TSRs were added to determine the effect on colonization levels at the onset of symbiosis. Overall, we describe a diverse TSR protein repertoire in anthozoans that contains homologues to known vertebrate proteins in addition to novel domain combinations. In addition, we provide functional evidence for the importance of host-derived TSR proteins in the establishment of the cnidarian-dinoflagellate symbiosis.

## Results

### Cnidarian TSR proteins

The overall numbers of TSR proteins identified from the four genomes, *N. vectensis*, *A. pallida*, *A. digitifera*, and *S. pistillata* were much higher than those identified from transcriptomes. Searches revealed a rich and diverse repertoire of TSR proteins within the seven anthozoan species, when compared to mammalian TSR superfamily members of known function; the largest groups identified were the ADAMTS metalloproteases and the properdin-like TSR-only proteins (*Figure 2*). Putative thrombospondins with similar domain structure to human thrombospondins 3, 4 and 5 were identified in all species. None of the cnidarian resources searched contained a thrombospondin-like protein with TSR domains. Large numbers of TSR-only proteins were identified in comparison to those known in mammals, where complement factor properdin is the only example of a protein containing only TSR repeats aside from a signal sequence. TSR protein sequences containing novel protein domain architecture were also identified, including those with astacin metalloproteases, von Willebrand factors (VWAs), trypsin, *Stichodactyla helianthus* K$^+$ channel toxin (ShK) domains and immunoglobulin domains (*Figure 2*).

### Analysis of potential binding sites and conserved motifs in cnidarian TSR domains

TSR domains taken from a selection of identified cnidarian TSR proteins, show very strong amino acid sequence homology to the second TSR repeat in the human thrombospondin 1 protein (*Figure 2—figure supplement 1*). Features contributing to the three-dimensional folded protein described from the crystal structure of the TSR repeat of human thrombospondin 1 (*Tan et al., 2002*) are present in the cnidarian TSRs, including: (1) six cysteine residues, shown to form disulfide bridges; (2) three tryptophan residues forming the WXXWXXW motif which participates in protein and glycosaminoglycan binding sites (GAG binding); and (3) polar residues (such as arginine, lysine and glutamine) present in the RXRXR motif, forming salt bridges with other polar residues that aid in folding. In addition, all sequences contain the CSVTCG and GVQTRXR motifs, which bind SRB1/CD36 (*Zhang and Lawler, 2007*).

### TSR proteins in *Symbiodinium minutum* and *S. microadriaticum*

Searches of the *S. minutum* genome identified 175 contigs containing TSR domains, however none of the predicted proteins contained VWA domains (*Figure 3*). TSRs were alone or in repeats of up to 16. In contrast, most apicomplexan TSR protein sequences possess one or more VWA domains

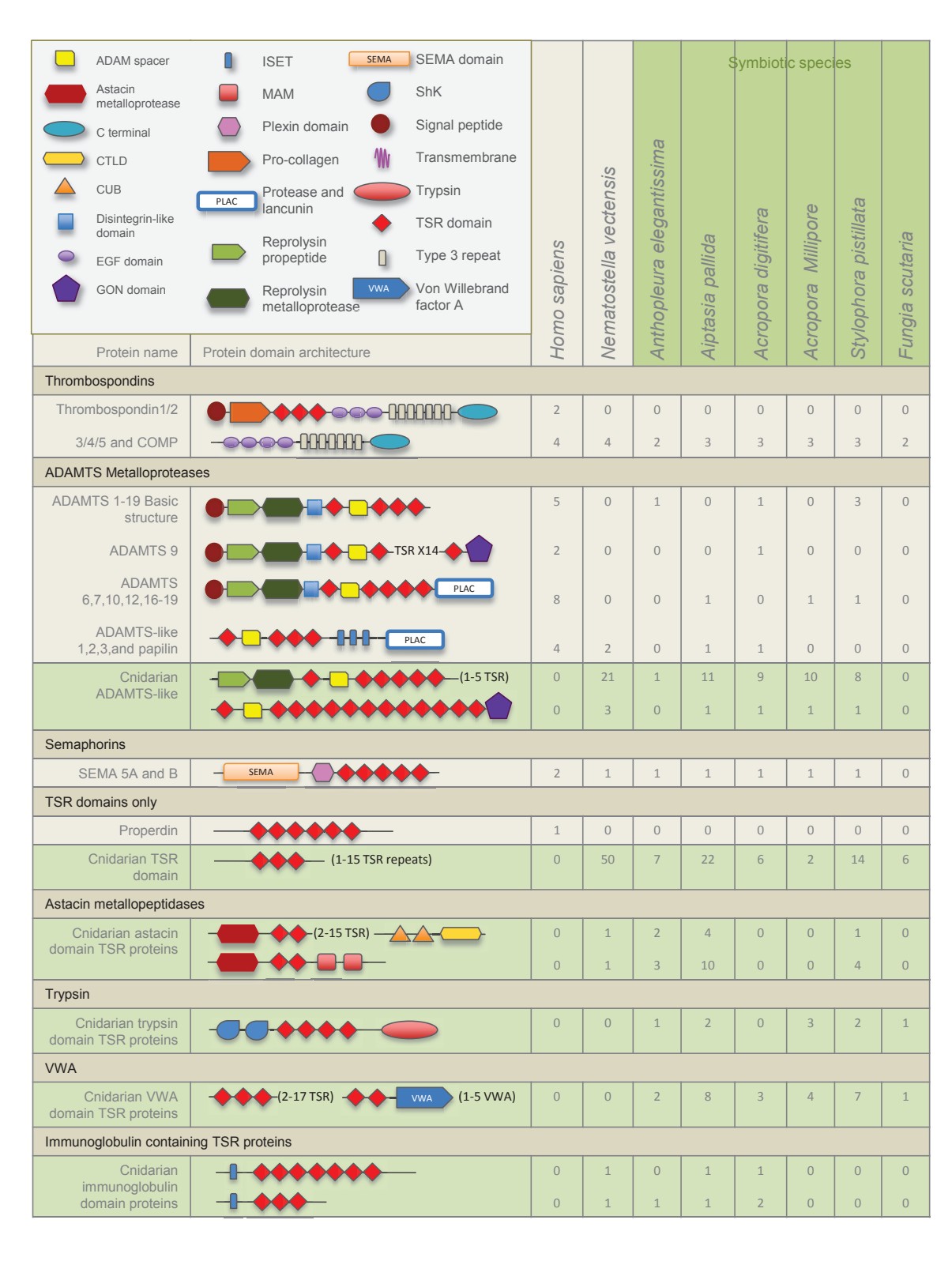

**Figure 2.** Domain architecture of cnidarian TSR super-family proteins compared to known vertebrate TSR-domain-containing proteins.
The following figure supplement is available for figure 2:

*Figure 2 continued on next page*

Figure 2 continued

**Figure supplement 1.** The TSR domain is very well conserved from cnidarians up to humans, with binding motifs for glycosaminoglycans (GAGs) and the type B scavenger receptors, CD36/SRB1.

and all have a C-terminal transmembrane domain. Searches of the *S. microadriaticum* genome revealed similar results and included proteins containing only the TSR domains in repeats up to 20. An alignment of TSR domains, including those from apicomplexan TRAP proteins, human thrombospondins 1 and 2, *S. minutum*, *S. microadriaticum* and two cnidarian TSR proteins is shown in *Figure 3—figure supplement 1*. *S. minutum* TSRs have five or six cysteines, a variation that is consistent with apicomplexan TRAP proteins (*Morahan et al., 2009*). The CD36/SRB1 but not the GAG-binding sites are well conserved in *S. minutum* sequences. In contrast, *S. microadriaticum* TSR domains contain six cysteines and are more similar to human and cnidarian TSRs than apicomplexan TSR domains.

## Evidence of TSR domain proteins in host but not symbiont

Anti-human TSR labelled two bands of 40 and 47 kDa in immunoblot analysis of homogenates from symbiotic *A. pallida* protein and a single band at 40 kDa in aposymbiotic *A. pallida* (*Figure 4A* and *Figure 4—figure supplement 1*). Immunofluorescence of *S. minutum* using anti-human TSR showed label on freshly isolated but not cultured cells. DiI lithophylic membrane stain labelled freshly isolated but not cultured *S. minutum* cells (*Figure 4—figure supplement 2*). Likewise, anti-TSR signal was absent from cultured *S. minutum* cells (*Figure 4B*) but appeared around the outside of freshly-isolated *S. minutum* cells (*Figure 4C*), suggesting that it labels the host symbiosome membrane complex and/or host material associated with the freshly isolated cells. Immunofluorescent labelling of symbiotic anemone tentacle cryosections showed antibody label in host gastrodermal tissue when in close association with resident symbionts (*Figure 4D,E*). Secondary antibody-only and IgG controls showed no labeling (*Figure 4F*).

## Blocking TSR domains inhibits symbiont uptake by host anemones

Incubation of aposymbiotic anemones with anti-human TSR prior to and during symbiont inoculation resulted in strong and statistically significant (mixed effects ANOVA $F_{(2, 24)}=16.55$, $p<0.0001$) inhibition of host colonization by *S. minutum* (*Figure 5A*). Levels of colonization stayed very low throughout the treatment period, rising to only $1.26 \pm 0.86\%$. In contrast, anemones incubated in both the FSW and IgG antibody controls showed moderate rates of colonization for the first 72 hr, but a dramatic increase thereafter to $18.1 \pm 2.65\%$ and $17.8 \pm 2.56\%$, respectively, by 120 hr post-inoculation.

## Addition of exogenous human thrombospondin-1 results in 'super colonization' of hosts by symbionts

Addition of exogenous human thrombospondin-1 protein increased the rate of host colonization by symbionts. Anemones pre-treated with thrombospondin-1 showed markedly increased (mixed effects ANOVA $F_{(1, 16)} = 59.36$, $p<0.0001$) colonization success compared to FSW controls (*Figure 5B*). Colonization success after 48 hr was $8.05 \pm 0.98\%$ in the thrombospondin-1 treatment compared to $1.18 \pm 0.28\%$ in the FSW treatment. By 96 hr post-inoculation, colonization success had risen to $25.1 \pm 2.6\%$ in the thrombosondin-1 treatment compared to just $9.87 \pm 2.4\%$ in the FSW control. By the end of the experiment, at 120 hr post-inoculation, colonization levels in control animals had almost caught up to those in treatment ones, suggesting that the stimulatory impact of thrombospondin-1 was most pronounced during the first 96 hr of symbiosis establishment.

## Addition of exogenous *A. pallida* TSR peptide fragments during inoculation increases colonization success

As with human thrombospondin-1, pre-treating anemones with short synthetic *A. pallida* TSR peptides resulted in increased colonization success (mixed effects ANOVA $F_{(2, 24)} = 69.46$, $p<0.0001$; *Figure 5C*). At 48 hr post-inoculation, symbiont levels were higher in anemones pre-treated with either peptide (Peptide 1: $11.14 \pm 1.1\%$; Peptide 2: $11.78 \pm 0.9\%$) compared to the FSW-only

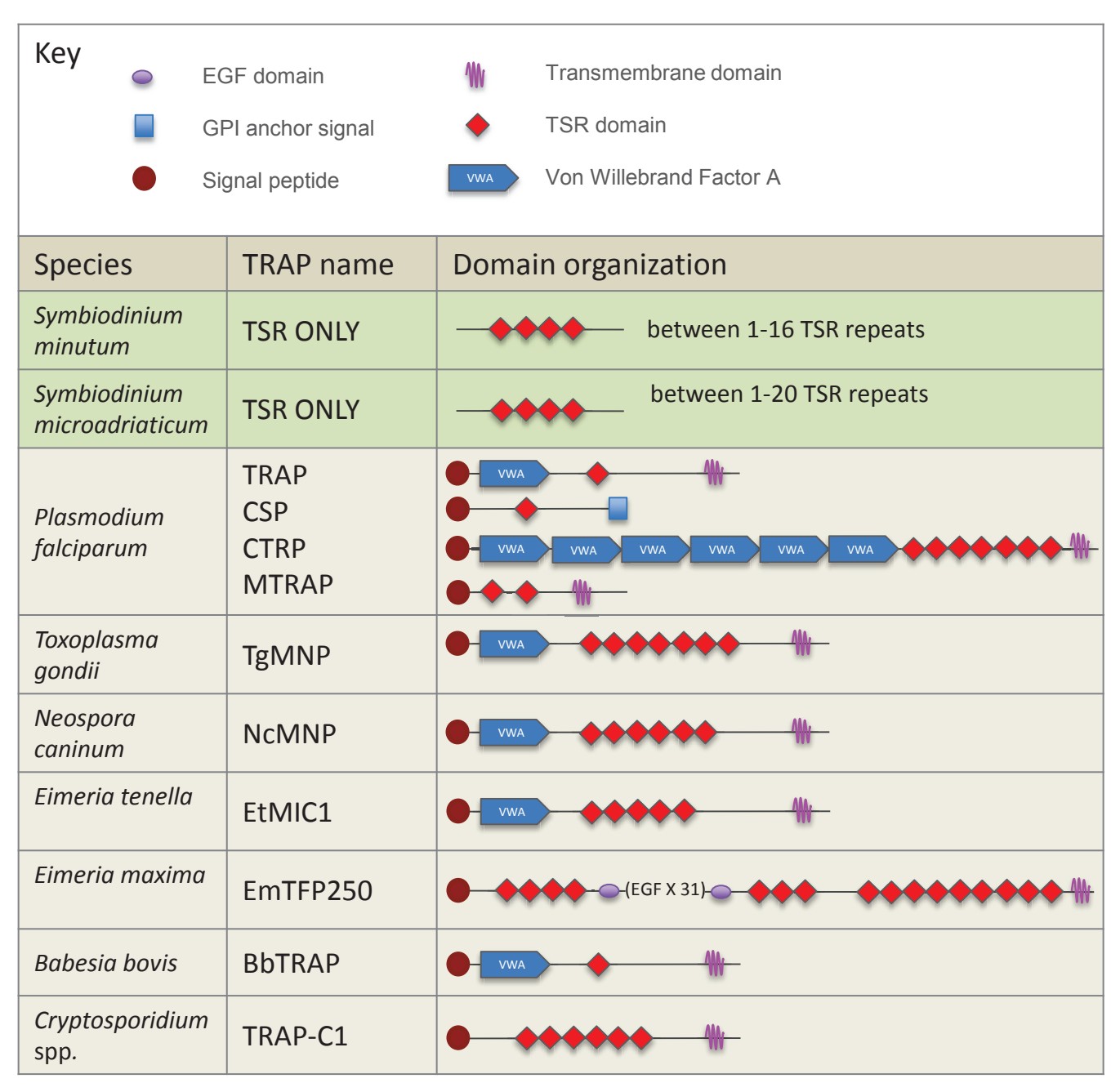

**Figure 3.** Schematic representation of members of the TSR gene family in dinoflagellates and apicomplexan parasites. TSRs from the dinoflagellates *Symbiodinium minutum* and *S. microadriaticum* are shown in green. Apicomplexan TRAP proteins are shown in beige.

The following figure supplement is available for figure 3:

**Figure supplement 1.** TSR domain alignment compares apicomplexan TRAP TSR domains with TSR domains from the dinoflagellates *Symbiodinium minutum* and *S. microadriaticum*, TSR 2 from human TSP1, and ADAMTS-like TSR domains from the anemones *Nematostella vectensis* and *Aiptasia pallida*.

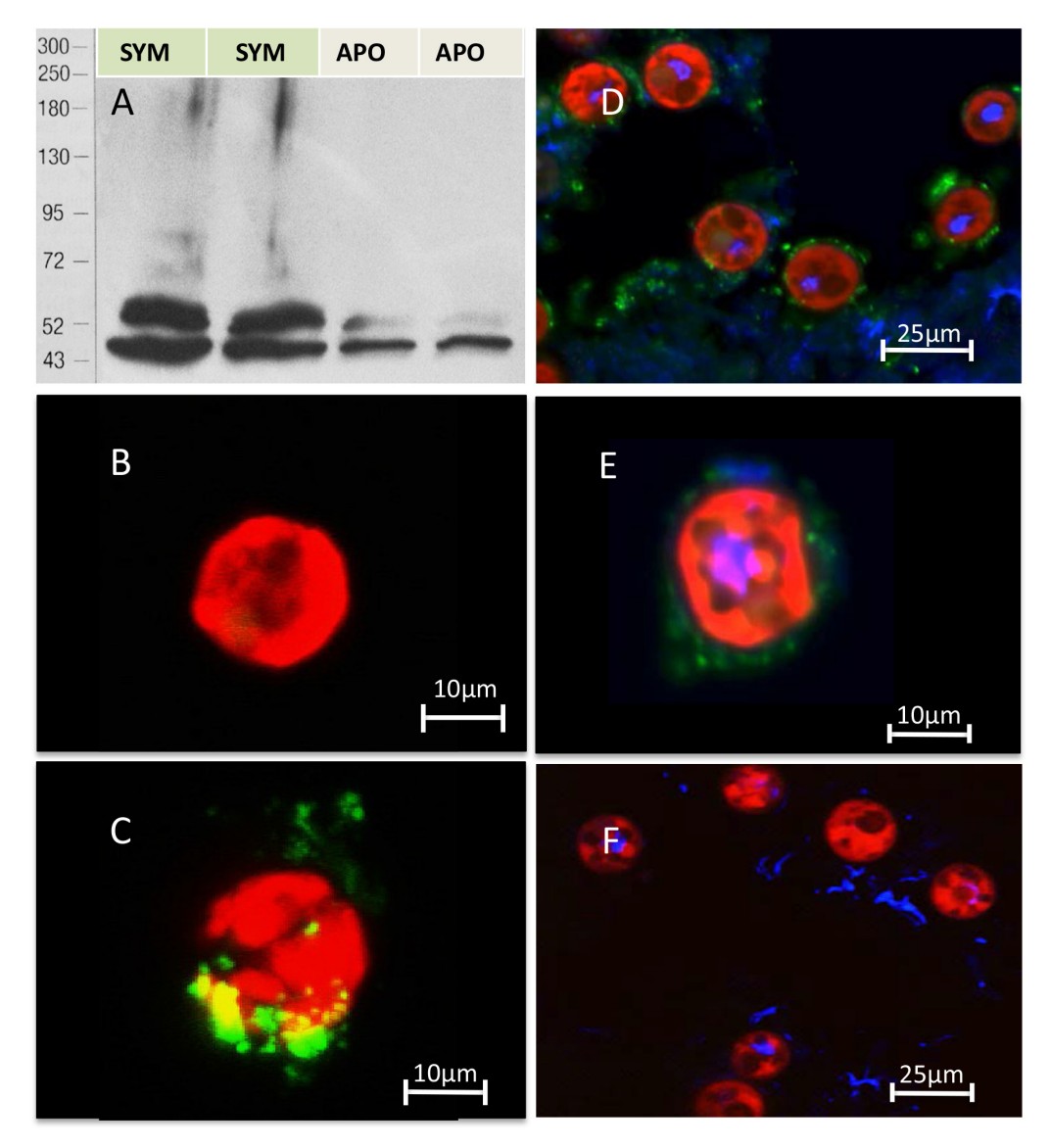

**Figure 4.** Immuno-analyses using anti-thrombospondin show evidence of TSRs in symbiotic anemone host tissues. (**A**) Immunoblots of symbiotic (SYM) and aposymbiotic (APO) *A. pallida* label bands at 40 and 47 kDa in symbiotic anemones and a single band at 40 kDa in aposymbiotic anemones. (**B, C**) Confocal images of dinoflagellate cells taken from (**B**) culture or (**C**) freshly isolated cells taken from *A. pallida* homogenates. A fluorescent probe conjugated to anti-human thrombospondin does not label cells from culture (**B**) but strongly labels host cell debris and/or membranes associated with freshly isolated cells (**C**). (**D, E**) Confocal images of cryosections from symbiotic *A. pallida* gastrodermal tissue stained with anti-thrombospondin at lower (**D**) and higher (**E**) magnification. Anti-thrombospondin labelling is evident in host tissues surrounding symbionts. (**F**) Confocal image of control anemone cryosections incubated with secondary antibody only. No anti-thrombospondin labeling is evident. Green = anti-thrombospondin, Red = algal autofluorescence, blue = DAPI stain of host and symbiont nuclei.

The following figure supplements are available for figure 4:

**Figure supplement 1.** A: Actin control for immunoblot blot in *Figure 4*.

**Figure supplement 2.** Lipophilic membrane staining of dinoflagellate cells using Dil.

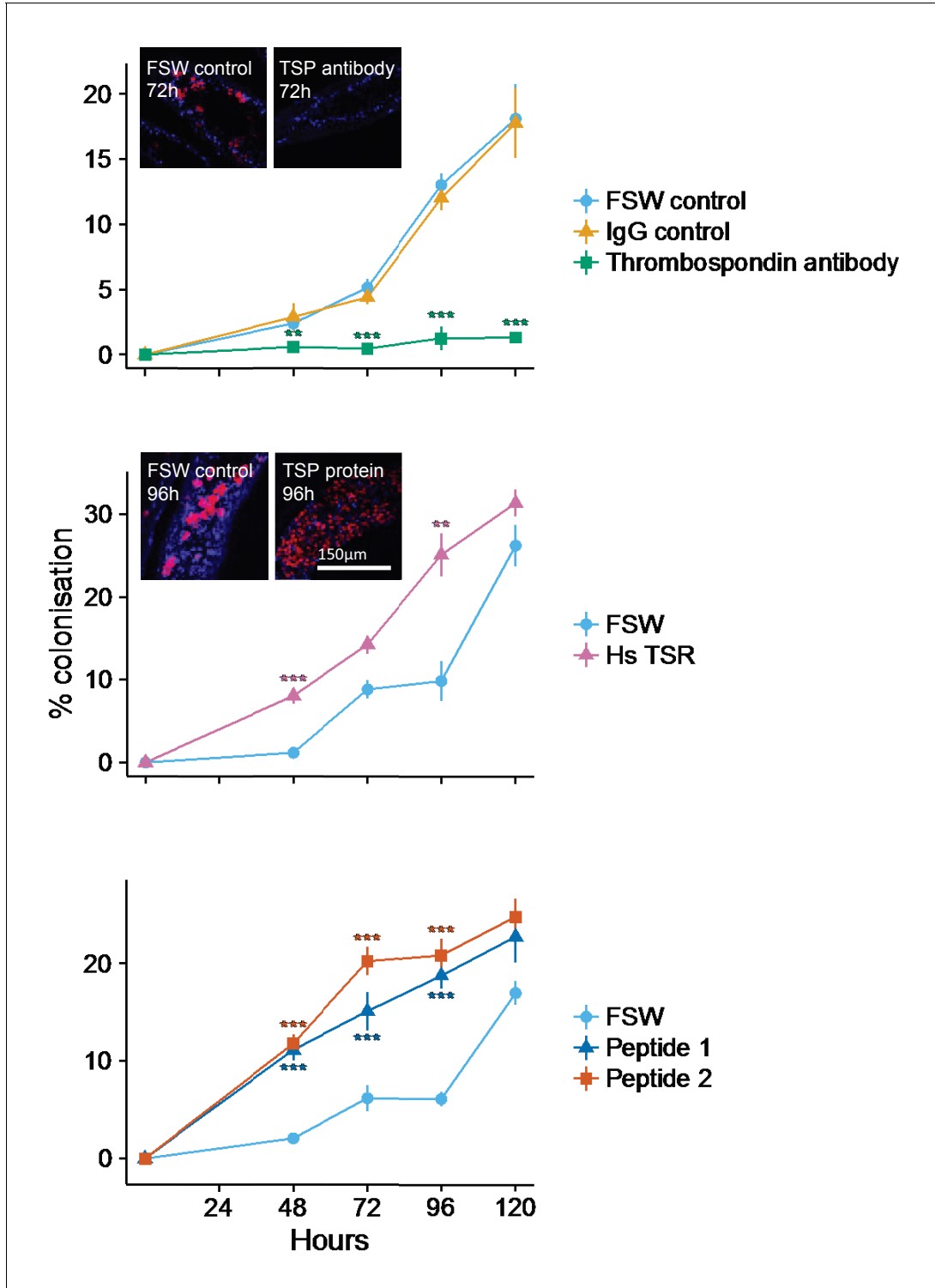

**Figure 5.** Kinetics of recolonization after antibody and peptide treatments. (**A**) Anemones pre-incubated in an anti-human thrombospondin (green) show decreased colonization success compared FSW-only (light blue) and IgG (orange) controls. Inset: confocal images show representative tentacle slices at 72 hr post-inoculation. (**B**) The addition of exogenous human thrombospondin-1 (purple) significantly increased the colonization rate during colonization, compared to control anemones in FSW (blue). Inset confocal images show representative tentacle slices at 96 hr post-inoculation. (**C**) The effect of synthetic TSR peptides 1 (blue) and 2 (orange) on colonization rates compared to the control anemones in FSW. Anemones treated with both peptides 1 and 2 showed increased uptake of algae during colonization. Statistical significance of treatment effects was assessed using mixed effects models, with contrasts calculated between individual treatments and FSW at each time-point; ***p<0.001; *p<0.05; p<0.1.

*Figure 5 continued on next page*

*Figure 5 continued*

The following source data is available for figure 5:

**Source data 1.** Source data used for statistical analyses described in results and depicted in *Figure 5A*: Long-form table with experimental results described in the results section *Blocking TSR domains inhibits symbiont uptake by host anemones* and shown in *Figure 5A*.
**Source data 2.** Summary statistics (mean and s.e.) displayed in *Figure 5A*.
**Source data 3.** Source data used for statistical analyses described in results and depicted in *Figure 5B*.
**Source data 4.** Summary statistics (mean and s.e.) displayed in *Figure 5B*.
**Source data 5.** Source data used for statistical analyses described in results and depicted in *Figure 5C*.
**Source data 6.** Summary statistics (mean and s.e.) displayed in *Figure 5C*.

controls (2.08 ± 0.29%). After 48 hr, colonization levels in the Peptide 2 treatment were consistently higher than in the Peptide 1 treatment. This difference was particularly apparent at 72 hr, where colonization levels in anemones in the Peptide 2 treatment were 5% higher than in Peptide 1 (20.2 ± 1.4% and 15.11 ± 1.98%, respectively). The peptide treatments showed the largest increase relative to the FSW control at 96 hr, with 18.8 ± 1.3% and 20.9 ± 1.68% colonization for Peptides 1 and 2, respectively, compared to only 6.15 ± 0.75% for the FSW control. However, as in the thrombospondin-1 treatment, by the end of the experiment at 120 hr, colonization in the control animals had reached levels similar to those in the peptide-treated anemones, suggesting once again that the impact of TSR peptides was most pronounced early in the colonization process.

## Ap_Sema-5 expression increases at early time-points during the onset of symbiosis

To investigate the specific TSR proteins involved in the onset of symbiosis, gene expression of two sequences obtained from the bioinformatics searches of the *A. pallida* genome was measured using quantitative PCR (qPCR). The first sequence, Ap_Sema5 (AIPGENE5874) has a domain structure similar to the vertebrate semaphorin-5 sequence with an N-terminal Sema domain and C-terminal TSR. This sequence was selected for further investigation due to its role in tumor cell motility and invasion through modifications to the actin cytoskeleton (*Li and Lee, 2010*), which suggests it could play a role in cytoskeletal rearrangements during symbiont uptake. The second sequence, Ap_Trypsin-like (similar to AIPGENE 1852), represents a novel domain combination as it possesses two N-terminal ShK domains, four TSR domains, and a C-terminal trypsin domain. The peptide used in the functional experiments described above was designed specifically to this sequence, therefore making it an interesting target for further investigation. Furthermore, in the genome searches, a similar sequence was found in symbiotic species, but not the non-symbiotic *Nematostella vectensis*, suggesting this protein may play a role in symbiosis. Quantitative PCR results revealed similar expression trends for both Ap_Sema5 and Ap_Trypsin during the onset of symbiosis (*Figure 6*). Ap_Sema5 showed a significant upregulation at 12 hr post-inoculation (estimate: −2.26, 95% c.i.: [−3.52; −1.01], p=0.0072) in the inoculated compared to aposymbiotic treatment, but by 72 hr post-inoculation it was significantly downregulated (estimate: 1.98, 95% c.i.: [0.73; 3.23], p=0.015). Ap_Trypsin-like displayed a downward trend in expression during the establishment of symbiosis (ANOVA, $F_{(1, 10)}$=5.90, p=0.036), however individual pairwise comparisons were not significantly different (see Supplementary *Source code 1* for detailed outputs of individual estimates and test statistics, including all pairwise comparisons at individual time-points).

## Discussion

### Bioinformatic searches reveal a diversity of anthozoan TSR proteins

Bioinformatic searches revealed a notable diversification of TSR-only proteins. This suggests that TSR proteins can be added to the growing list of immunity genes in cnidarians that are greatly

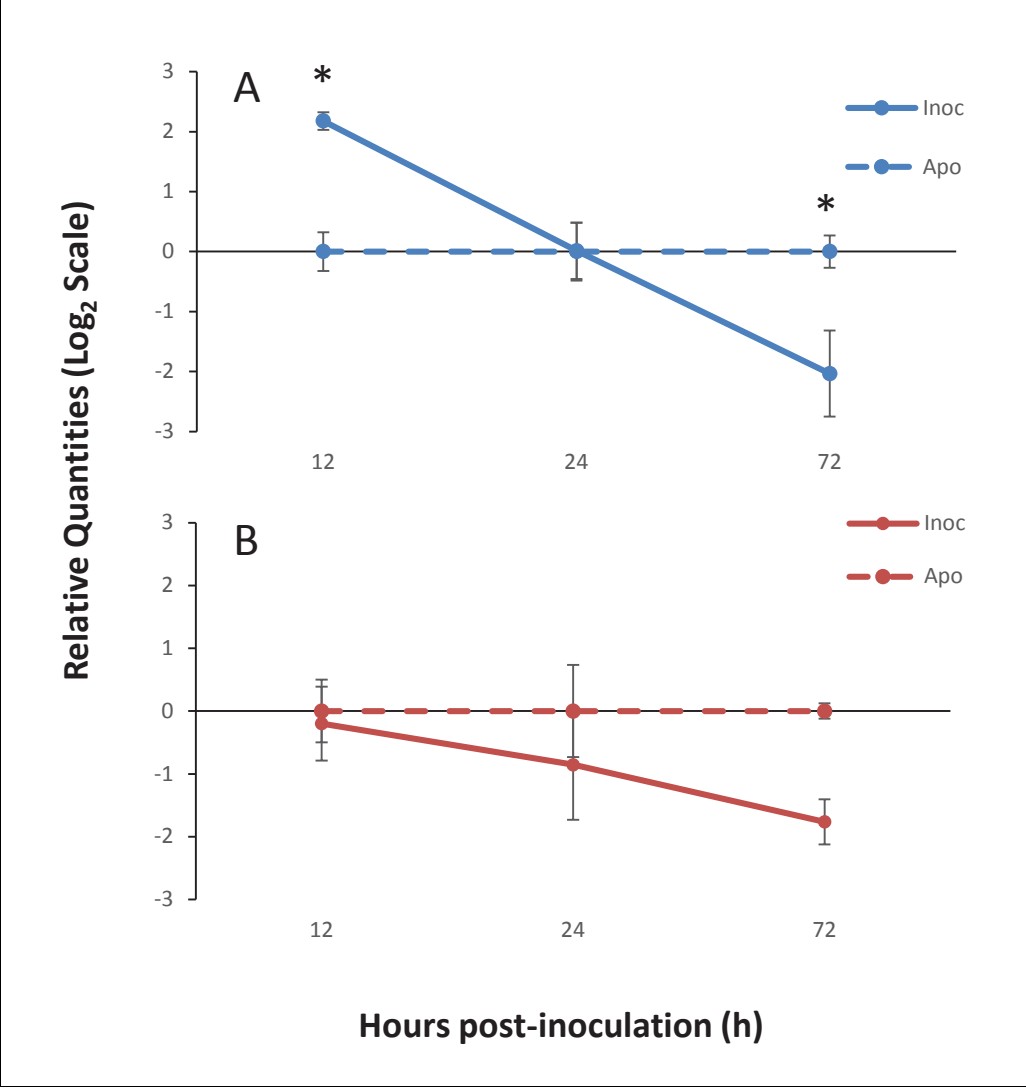

**Figure 6.** Gene expression of Ap_Sema5 and Ap_Trypsin-like at the onset of symbiosis. The relative quantities from qPCR on the log$_2$ scale are shown for animals that were inoculated with symbionts ('Inoc'; solid line) and those that remained aposymbiotic ('Apo'; dashed line). Bars represent means ± SE (n = 3) and stars represent significantly different levels of expression between the inoc and apo treatments at a particular time point (two-way ANOVA, Tukey's *post hoc* test). *p<0.05, **p<0.01.
The following source data is available for figure 6:

**Source data 1.** Source data used for statistical analyses described in results and depicted in *Figure 6*.

diversified compared to their counterparts in vertebrate genomes. These include expansions of toll-like receptors in *A. digitifera*, ficolin-like proteins in *A. pallida*, NOD-like receptors in *Hydra magnipapillata* and *A. digitifera*, and scavenger receptors in a variety of cnidarians (*Lange et al., 2011*; *Shinzato et al., 2011*; *Hamada et al., 2013*; *Baumgarten et al., 2015*; *Neubauer et al., 2016*). It has been hypothesized that such an expanded repertoire in basal metazoans is an alternate evolutionary strategy to vertebrate adaptive immunity that would enable complex reactions to, and management of, their microbiomes (*Hamada et al., 2013*).

The TSR-only repertoire expansion is of particular interest because these sequences are similar to the vertebrate complement protein properdin. This protein is known to have two interrelated functions that may be of particular relevance to the establishment of the cnidarian-dinoflagellate

symbiosis. First, properdin can act as a pattern recognition receptor (PRR), detecting microbe-associated molecular patterns (MAMPs) on invading microbes and triggering phagocytosis of microbes directly. Secondly, it can participate in the complement system alternative pathway, where it activates and stabilizes the proteolytic C3 convertase complex, which attaches to the surface of invading microbes and hence marks them for phagocytosis and/or lysis (*Hourcade, 2006*; *Spitzer et al., 2007*). There is growing functional evidence that the complement system, which is classically thought to function in defense against pathogens, also plays a role in the onset and regulation of cnidarian-dinoflagellate symbiosis (*Kvennefors et al., 2008*, *2010*; *Baumgarten et al., 2015*; *Poole et al., 2016*). Therefore, a testable hypothesis is that cnidarian TSR-only proteins function in a similar manner to vertebrate properdin, as either a PRR to recognize *Symbiodinium* or to interact with complement proteins to promote phagocytosis of symbionts. A recent transcriptomic study indicated that there was decreased expression of a transmembrane domain-containing-TSR-only protein in *A. pallida* larvae during the later stages of symbiosis establishment when compared to aposymbiotic larvae (*Wolfowicz et al., 2016*). Therefore, it is possible that this protein may serve as a PRR, with a high level of expression in aposymbiotic larvae and during inter-partner surface recognition, but decreased expression after phagocytosis.

We characterized a large repertoire of cnidarian ADAMTS metalloprotease-like proteins. The TSR domains within these cnidarian proteins are highly conserved and functional motifs are intact, including the tryptophan glycosaminoglycan-binding (GAG) motif 'WXXW', and scavenger receptor binding motifs 'CSVTCG' and 'GVITRIR' (*Adams and Tucker, 2000*; *Silverstein, 2002*). In humans, the metalloprotease ADAMTS 13 binds to SRB1 (*Davis et al., 2009*), and in *C. elegans* an ADAMTS protein (AD-2) is responsible for initiating the TGF$\beta$ pathway, regulating body growth and maintaining cuticle formation (*Fernando et al., 2011*). It is therefore conceivable that an ADAMTS-like TSR protein is involved in promoting tolerance in the cnidarian-dinoflagellate symbiosis.

TSR proteins containing the trypsin domain, ShK domain and the VWA domain, were present in five of the six symbiotic cnidarians, the trypsin containing TSRs identified in *A. digitifera* lack the ShK domain (*Figure 2*). The ShK domain is found in peptides that function as potassium channel inhibitors and it has been proposed that proteins that include ShK in combination with other domains, such as trypsin, may also modulate channel activity (*Rangaraju et al., 2010*). Additionally, proteins with ShK or TSR domains have previously been found in nematocysts (*Balasubramanian et al., 2012*; *Rachamim et al., 20142015*). Therefore, ShK plus trypsin proteins are likely toxin proteins that function in nematocysts and food acquisition. Interestingly, qPCR results indicated that Ap_Trypsin-like has a trend of decreased expression during the establishment of the symbiosis (*Figure 6*). Therefore, it is still unclear what role this protein plays in symbiosis. This downward trend could indicate a de-emphasis by the host on food capture, as it transitions to gaining nutritional support from its symbionts. A more detailed comparative study would need to be performed to determine whether these sequences are truly differentially distributed as a function of symbiosis.

The comprehensive search for TSR-containing thrombospondin homologues found no sequences in any of the anthozoan resources examined (*Figure 2*). This strongly suggests that TSR-containing thrombospondins are not present in cnidarians. However, searches for TSR proteins within anthozoans revealed a rich diversity of TSR superfamily members, including some whose domain architectures bear a strong resemblance to members in other animals and others with novel domain architectures. Domain abundance and architecture show no clear pattern based on symbiotic state or anthozoan phylogeny, but instead correlate to type of resource searched: genomes provide better representation of TSR abundance than transcriptomes. It is likely that a more accurate picture of TSR protein diversity will emerge over time as more genomes become available and annotations improve.

## *Symbiodinium* TSR proteins show limited similarities to apicomplexan TRAPs

Searches of both *S. minutum* and *S. microadriaticum* genomes revealed evidence of TSR proteins, but none that had all of the hallmarks of the TRAP proteins in apicomplexans. *Symbiodinium* TSR sequences contain a signal peptide and multiple TSR repeats, but not the VWA or transmembrane domains found in most apicomplexan TRAPs (*Figure 3*). It is therefore unlikely that *Symbiodinium* is using TSR proteins to attach to hosts *via* mechanisms homologous to those used by apicomplexans. Expression profiles and localization studies of symbiont TSR proteins in culture *vs. in hospite* could

provide insight into whether these proteins are playing a role in the symbiosis. Interestingly, the number of cysteines contained in the TSR domains differed between the two species. *S. minutum* TSR domains contained five cysteines, similar to apicomplexan TSRs. In contrast, *S. microadriaticum* TSRs contained six, similar to metazoan TSRs.

## Colonization experiments implicate the TSR domain in symbiosis establishment

We introduced dinoflagellates to aposymbiotic anemones that had been pre-treated to either block or mimic TSR proteins. Blocking TSR domain function resulted in colonization levels reduced to 1% infection and below, providing strong evidence for the involvement of TSR proteins in the establishment of the symbiosis. The anti-human TSR epitope corresponds to three TSR repeats and is therefore indiscriminate in its blocking effect of TSR proteins. Results indicate a role for host, rather than symbiont TSR proteins in symbiosis establishment, given the localization of anti-thrombospondin to host tissues, including those of aposymbiotic anemones, and not the outer surface of cultured *Symbiodinium* cells (*Figure 4*).

Treatment of *A. pallida* with exogenous TSR domains provided further evidence for the role of host TSR proteins in the early onset of the symbiosis. Due to high levels of TSR domain conservation across taxa, synthetic peptides designed from TSR domains have been employed by a number of studies, including determining which motifs bind to CD36 (*Li et al., 1993*) and which *Plasmodium* TSR peptides bind to red blood cells (*Calderón et al., 2008*). The synthetic peptide used in this study contained both the tryptophan GAG-binding motif 'WXXW', and scavenger receptor binding motifs 'CSVTCG' and 'GVXTRXR'. This result suggests that one or multiples of these binding motifs are involved in successful entry to host cells by the dinoflagellates.

Treatment of *A. pallida* with human thrombospondin and synthetic *A. pallida* TSR peptides resulted in 'super colonization' by the symbionts (*Figure 5B,C*). These results provide evidence against the hypothesis of membrane-linked host TSRs serving as PRRs to promote inter-partner recognition. We suggest that exogenous TSRs would compete with membrane bound host TSR PRRs for *Symbiodinium* MAMPs, and result in decreased colonization success. Instead, our results support a hypothesis of TSRs enhancing symbiont colonization through steric interactions with a secondary molecule(s), be it C3 convertase complex, SRB1, or some other protein that promotes phagocytosis. In this case, addition of exogenous TSRs would result in binding of additional secondary proteins that would in turn promote phagocytosis and result in the 'super colonization' observed. This hypothesis is further supported by sequence data which indicate that the majority of cnidarian TSR proteins lack transmembrane domains (see *Supplementary file 1*).

Our initial interest in the TSR domain was prompted by the search for a binding target for the host cell scavenger receptor SRB1, which is upregulated in the symbiotic state of *A. pallida* and another sea anemone, *A. elegantissima* (*Rodriguez-Lanetty et al., 2006*; *Lehnert et al., 2014*). In other systems, SRB1-TSR interactions are implicated in promoting phagocytosis and initiating the tolerance promoting TGF$\beta$ pathway by activating latent TGF$\beta$ protein (*Khalil, 1999*; *Murphy-Ullrich and Poczatek, 2000*; *Koli et al., 2001*). The addition of TSR protein may have dual functions, firstly to enhance phagocytosis of microbes and secondly to promote tolerance. Many intracellular parasites manipulate host innate immune defence mechanisms to their own advantage (*McGuinness et al., 2003*).

Gene expression results also provide evidence of a role for TSR proteins at the onset of symbiosis (*Figure 6*). Ap_Sema5 showed increased expression at early time points during onset of symbiosis, but decreased expression at later time points, indicating that it may play a role in initial recognition and uptake of symbionts, but not subsequent proliferation. Future experiments that target earlier time points during the onset of symbiosis could provide evidence to support this hypothesis. Interestingly, the decreased trend in expression at 72 hr post-inoculation is similar to the downregulation observed for several TSR protein genes and a non-TSR semaphorin (Semaphorin-3E) in symbiotic *A. pallida* larvae five to six days post-inoculation (*Wolfowicz et al., 2016*). Due to the pleiotropic nature of semaphorins, further investigation of the precise role of Ap_Sema5 is needed. Intriguingly, however, vertebrate semaphorin-5a has been shown to play a role in modifications to the actin cytoskeleton, and it therefore could function in the phagocytosis of symbionts (*Li and Lee, 2010*). Moreover, semaphorin-5a has been shown to promote cell proliferation and to inhibit apoptosis in several cancers (*Sugimoto et al., 2006*; *Pan et al., 2010*; *Sadanandam et al., 2010*), raising the possibility

that it could promote immunotolerance of foreign *Symbiodinium* cells. Lastly, Ap_Sema5 could function as a PRR. In vertebrates, Semaphorin-7a has previously been shown to serve as an erythrocyte receptor for a *Plasmodium* TRAP protein (*Bartholdson et al., 2012*), where the sema domain of semaphorin-7a interacts with a TSR domain in the TRAP protein, to promote invasion of host red blood cells by the parasite. Overall, there are a variety of roles that Ap_Sema5 may play to promote the onset of symbiosis, and future functional experiments can be used to test these.

### Concluding remarks

Characterization of TSR proteins in cnidarians in this study has revealed a diverse repertoire of genes whose functions remain to be fully described. Functional work provides another piece in the complex web of inter-partner signaling that supports symbiont acquisition and presents the TSR as a protein domain potentially involved in nurturing positive microbial-host interactions in the cnidarian-dinoflagellate symbiosis. Studies using antibodies, proteins, peptides and qPCR to explore TSR protein function in symbiosis suggest that one or more host-derived TSR proteins is participating in host-symbiont communication.

Taken together, these studies point to these proteins, potentially working in concert with other secondary proteins, promoting phagocytosis of symbionts and enhancing colonization success. *Figure 7* presents a model summarizing the evidence emerging from the immunolocalization and functional experiments. Future studies should target specific TSR homologues for further investigation using antibodies made against specific proteins and ideally using knockdown or gene-editing technologies that would empirically test the impact of these genes on host-symbiont recognition. Overall, there is mounting evidence that *Symbiodinium* cells can manipulate the host's immune defenses to gain entry to, and proliferate in cnidarian cells, as occurs in parasitic infections, but how these various strands of evidence ultimately tie together is still unclear and requires further investigation.

## Materials and methods

### Genomic and transcriptomic resources

To characterize the TSR protein repertoire in cnidarians, seven species with publically available resources were searched. These resources were selected to capture a diversity of anthozoans, with representatives from Actinaria, and the complex and robust clades of the scleractinains. Additionally, species were chosen to represent a variety of symbiotic states and symbiont transmission mechanisms. These included three anemone species: *A. elegantissima* (*Kitchen et al., 2015*), *A. pallida* (*Lehnert et al., 2012*; *Baumgarten et al., 2015*) and *Nematostella vectensis* (*Putnam et al., 2007*), and four coral species: *Acropora digitifera* (*Shinzato et al., 2011*), *Acropora millepora* (*Moya et al., 2012*), *Fungia scutaria* (*Kitchen et al., 2015*) and *Stylophora pistillata* (Voolstra et al., submitted). These resources were derived from various developmental stages and symbiotic states (*Table 1*). All resources were used without manipulation, with the exception of the *A. pallida* transcriptome, for which raw Illumina sequence reads for accession SRR696721 were downloaded from the sequence read archive (RRID:SCR_004891) entry (http://www.ncbi.nlm.nih.gov/sra/SRX231866) and reassembled using Trinity (RRID:SCR_013048, *Grabherr et al., 2011*). In addition, the genomes of the symbiotic dinoflagellates *Symbiodinium minutum* (ITS2 type B1) (*Shoguchi et al., 2013*) and *S. microadriaticum* (*Aranda et al., 2016*) were searched for TSR proteins, to investigate the presence of a potential TRAP-like protein.

### TSR sequence searching

To search for cnidarian TSR proteins, databases were queried using several search strategies to ensure that all sequences were recovered. BLASTp or tBLASTn searches with the second TSR domain from mouse and human thrombospondin-1 protein sequences, and the consensus sequence (smart00209: TSP1) from the conserved domain database (RRID:SCR_002077, http://www.ncbi.nlm.nih.gov/cdd) as queries were performed for each resource. Keyword searches using the terms TSP1, thrombospondin, ADAMTS, ADAM and SEMA were also performed where genome browsers allowed keyword searches of GO, KEGG and PFAM annotations. Lastly, representative *N. vectensis* sequences of each protein type (ADAMTS-like, SEMA, TRYPSIN and TSR-only) were also used as queries for tBLASTn searches of the other six cnidarian resources. A high e-value cutoff ($1 \times 10^{-1}$)

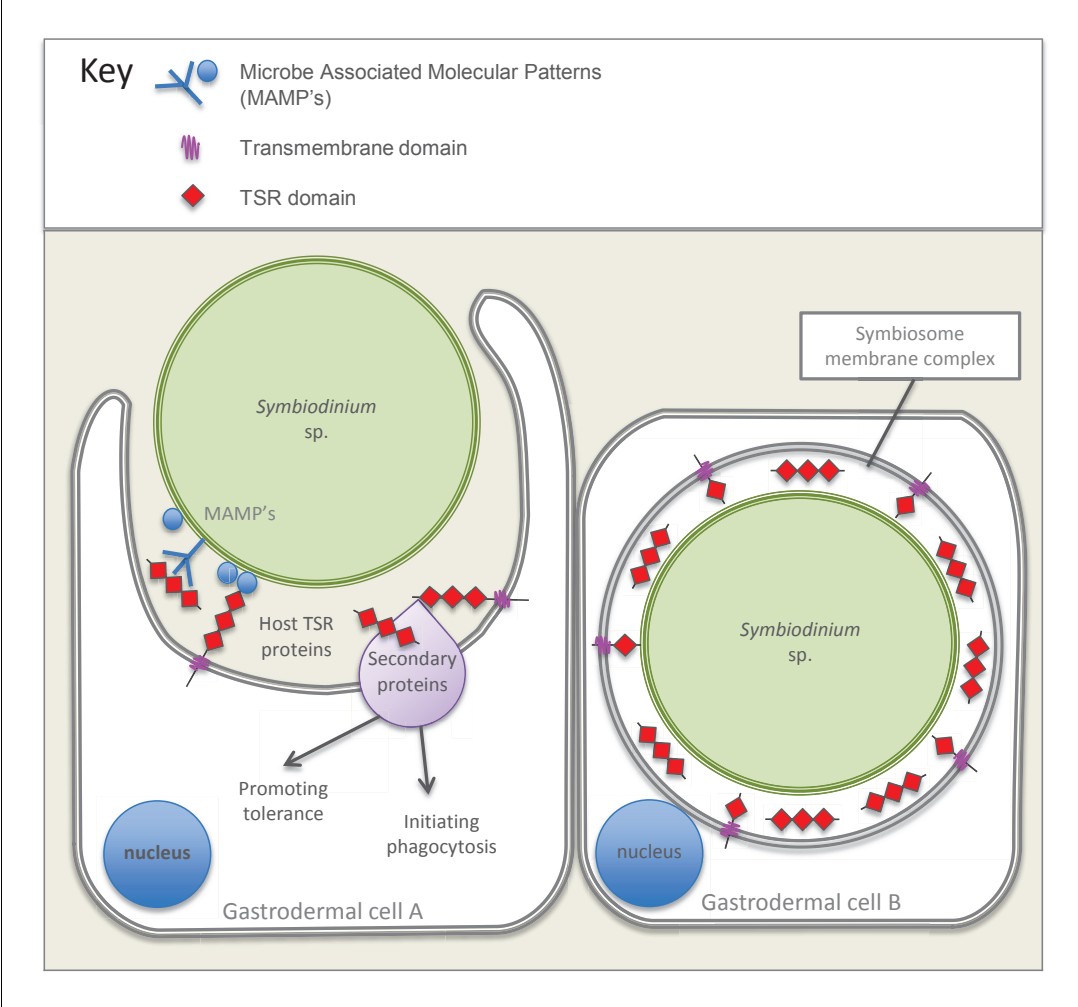

**Figure 7.** Model summarizing the evidence emerging from immunolocalization and functional experiments. Gastrodermal cell A depicts an aposymbiotic host cell in the process of symbiont acquisition. Results indicate that the addition of soluble TSR proteins promotes and enhances symbiont colonization. We suggest that secreted host TSR proteins may interact with MAMPs and/or secondary proteins to promote tolerance and initiate phagocytosis. Peptide experiments provide evidence against the hypothesis that membrane-linked host TSRs are serving as PRRs to promote inter-partner recognition; we hypothesize that host TSR proteins are secreted rather than membrane-anchored (see discussion text for further explanation). Gastrodermal cell B depicts a symbiotic host cell. Fluorescence microscopy suggests that TSR proteins are expressed within the host-derived symbiosome membrane complex and are concentrated around the symbionts within host gastrodermal tissue.

was used in the BLAST searches to recover divergent sequences. All BLAST searches were performed using Geneious pro version 7.1.8 (RRID:SCR_010519, *Kearse et al., 2012*) with the exception of the *N. vectensis*, *A. pallida* and *S. pistillata* genomes, for which searches were performed through the Joint Genome Institute online portal (RRID:SCR_002383), NCBI (RRID:SCR_004870) and the Reefgenomics online repository (RRID: SCR_015009, http://reefgenomics.org)(*Liew et al., 2016*), respectively. A list of metazoan resources searched is provided in *Table 1*. Sequences identified are tabulated in *Supplementary file 1*.

To confirm that the sequences obtained contained TSR domains, nucleotide sequences were translated using Geneious or ExPASy translate tool (RRID:SCR_012880, http://web.expasy.org/translate/) and then annotated using the Geneious InterProScan plugin (RRID:SCR_010519, *Kearse et al., 2012*). All annotations were double checked using the online protein domain database PfamA (RRID:SCR_004726, http://pfam.sanger.ac.uk), and only sequences that showed significant PfamA matches to a TSR domain with an e-value of $<1\times10^{-4}$ were used. Sequences for each species were aligned and those that were identical or almost identical (<5 aa difference in the conserved domains)

**Table 1.** Anthozoan and Dinoflagellate resources

| Organism | Family | Developmental stage | Symbiotic state | Data type | Publication |
|---|---|---|---|---|---|
| *Nematostella vectensis* | Edwardsiidae | Larvae | Non-symbiotic | Genome | *Putnam et al. (2007)* |
| *Anthopleura elegantissima* | Actiniidae | Adult | Aposymbiotic | Transcriptome | *Kitchen et al., 2015* |
| *Aiptasia pallida* | Aiptasiidae | Adult | Aposymbiotic | Transcriptome | *Lehnert et al. (2012)* |
| *Aiptasia pallida* | Aiptasiidae | Adult | Symbiotic | Genome | *Baumgarten et al. (2015)* |
| *Acropora digitifera* | Acroporidae | Sperm | Symbiotic | Genome | *Shinzato et al. (2011)* |
| *Acropora millepora* | Acroporidae | Adult and Larvae | Symbiotic | Transcriptome | *Moya et al. (2012)* |
| *Fungia scutaria* | Fungiidae | Larvae | Aposymbiotic | Transcriptome | *Kitchen et al., 2015* |
| *Stylophora pistillata* | Pocilloporidae | Adult | Symbiotic | Genome | Voolstra et al., submitted |
| *Symbiodinium minutum* | Symbiodiniaceae | culture ID Mf1.05b.01 | Dinoflagellate culture | Genome | *Shoguchi et al. (2013)* |
| *Symbiodinium microadriaticum* | Symbiodiniaceae | strain CCMP2467 | Dinoflagellate culture | Genome | *Aranda et al. (2016)* |

were omitted from the analysis, as they likely represented artefacts of assembly or different isoforms of the same protein. Sequences missing a start or stop codon were removed from the analysis. Diagrammatic representations of the protein domain configurations were produced using this information. Protein domain architectures were grouped together according to common domains and compared to known human TSR proteins (*Figure 2*).

## Maintenance and preparation of anemone and dinoflagellate cultures

A population (not necessarily clonal) of *Symbiodinium minutum* (clade B1)-containing *A. pallida*, originating from a local pet store, was maintained in saltwater aquaria at 26°C at a light intensity of approximately 40 µmol quanta $m^{-2}$ $s^{-1}$ with a 12/12 hr light/dark photoperiod, and fed twice weekly with live brine shrimp nauplii. Animals were rendered aposymbiotic by incubation for 8 hr at 4°C twice weekly for six weeks, followed by maintenance in the dark for approximately one month. Anemones were fed twice weekly with brine shrimp, and cleaned of expelled symbionts and food debris regularly.

Cultured dinoflagellates - *Symbiodinium minutum* (sub-clade B1; culture ID CCMP830 from Bigelow National Center for Marine Algae and Microbiota) - were maintained in 50 ml flasks in sterile Guillard's f/2 enriched seawater culture medium (Sigma, St. Louis, MO, USA). Dinoflagellate cultures were maintained at 26°C and 70 µmol quanta $m^{-2}$ $s^{-1}$ with a 12/12 hr light/dark photoperiod. CCMP830 cultures were typed using Internal transcribed spacer 2 (ITS2) sequencing in 2009 and 2016 to authenticate the identity of the culture. The CCMP830 cultures were not axenic and therefore *Mycoplasma* contamination testing was not performed.

In preparation for experimental manipulation, individual anemones were placed in 24-well plates in 2.5 ml of 1 µm-filtered seawater (FSW) and acclimated for 3–4 days, with the FSW replaced daily. Well-plates containing aposymbiotic anemones were kept at 26°C in the dark, while those containing symbiotic anemones were maintained in an incubator at a light intensity of approximately 40 µmol quanta $m^{-2}$ $s^{-1}$ with a 12/12 hr light/dark photoperiod. Animals were not fed during the acclimation or experimental periods.

## Immunoblot analysis of anti-thrombospondin protein targets

Immunoblots were performed on *A. pallida* proteins using an anti-human thrombospondin rabbit polyclonal antibody. The thrombospondin antibody was made against an epitope corresponding to the three TSR domains of human thrombospondin proteins 1 and 2 (Santa Cruz Biotechnology Cat# sc-14013 RRID:AB_2201952). The epitope showed sequence similarity to a TSR protein identified in *A. pallida* (*Figure 8A*). Groups of eight aposymbiotic or symbiotic anemones were homogenized on ice in 1 ml homogenization buffer (50 mM Tris–HCl, pH 7.4, 300 mM NaCl, 5 mM EDTA) with a protease inhibitor cocktail (BD Biosciences, San Jose, CA, USA). Homogenates were centrifuged at 4°C for 15 min at 14,000 x g to pellet cell debris, supernatants were decanted and protein concentrations were determined using the Bradford assay. Protein was adjusted or diluted in RIPA buffer to a

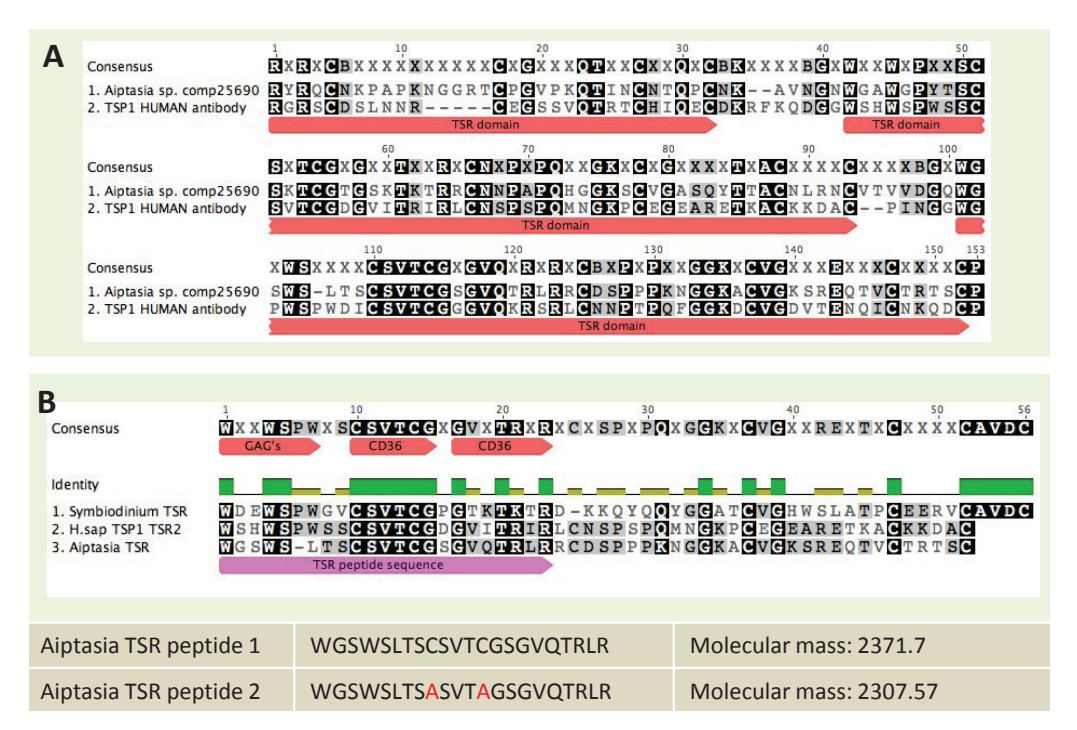

**Figure 8.** Sequence information for thrombospondin antibody and TSR peptide fragments used in this study. (A) Alignment of the second TSR domains from human thrombospondin 1 and TSR proteins from the anemone *Aiptasia pallida* and the dinoflagellate *Symbiodinium minutum.* In red are the binding sites for glycosaminoglycans (GAGs) and CD36; greyscale indicates the % identity of the three sequences. Pink annotation indicates the TSR peptide sequence covering all three binding domains; inset are the synthetic peptide sequences for experimental peptides. In Peptide 2, the cysteine residues were replaced with alanine residues, as shown in red. (B) A section of the antibody-binding region of the human thrombospondin 1/2 antibody (H-300, sc-14013 from Santa Cruz Biotechnology), aligned to a TSR protein fragment from *Aiptasia* sp. Legends for Supplementary Material.

standard concentration of 50 µg total protein *per* well and boiled for 5 min in loading dye. Proteins were resolved on a 7% SDS–PAGE gel and then electrophoretically transferred overnight onto nitrocellulose membrane. After blocking with 5% non-fat dry milk in TBS-Tween 20 (0.1%) for 1 hr at 37°C, membranes were incubated with anti-thrombospondin or an IgG isotype control, both at a dilution of 1:200, for 2 hr at room temperature. The blots were washed three times in TBS-Tween 20 followed by incubation in a HRP-conjugate goat anti-rabbit IgG Alexa Fluor 546 secondary antibody (Molecular Probes Cat# A-11030 RRID:AB_144695) at a 1:5000 dilution (0.2 µg ml$^{-1}$; Sigma, St. Louis, MO, USA) for 1 hr. Bands were detected by enhanced chemiluminescence (Millipore, Temecula, CA, USA). Blots were stripped and re-probed with an actin loading control (Santa Cruz Biotechnology Cat# Sc-1616 RRID:AB_630836), see *Figure 4—figure supplement 1* for actin control.

## Cryosectioning and immunofluorescence microscopy to localize TSR proteins

Immunofluorescence was used to investigate the presence of TSR proteins on the surface of dinoflagellate cells. We compared anti-human TSR binding in cultured *S. minutum* strain CCMP830 to *S. minutum* cells freshly isolated from *A. pallida.* To obtain freshly-isolated symbiont cells with intact symbiosome membranes, anemones were homogenized in a microfuge tube with a micro-pestle and the resulting homogenate was centrifuged at a low speed (<1000 rpm) for 5 min to produce an algal pellet. The pellet was washed several times in FSW and re-pelleted. Algal cells were re-suspended to a final concentration of $2.5 \times 10^4$ cells *per* ml. The lipophilic membrane stain, DiI (1,1'-dioctadecyl-3,3,3',3'-tetramethylindocarbocyanine perchlorate, DilC18(3); Molecular Probes), was used to test for the presence of putative symbiosome membrane surrounding freshly isolated symbiont cells and cells taken from culture. DiI was added to cells in 500 µl of FSW in a microfuge tube and gently

mixed shortly before small amounts of suspended cells were placed on a well slide and imaged. Both cultured and freshly isolated *S. minutum* cells were incubated in the anti-human TSR conjugated to the secondary antibody Alexa Fluor 546 goat anti rabbit IgG fluorescent probe (Molecular Probes Cat# A-11030 RRID:AB_144695) at a 1:1000 dilution. Anti-thrombospondin and DiI labeling in cells was imaged using a Zeiss LSM 510 Meta microscope through a Plan- APOCHROMAT 63x/1.4 Oil DIC objective lens. See *Supplementary file 2* for a description of fluorescent dyes and the specific excitation and emission wavelengths.

To localize TSR proteins in symbiotic and aposymbiotic anemone tissues, cryosections of anemone tentacles were made using methods modified from *Dunn et al. (2007)*. The sections were washed twice in PBS and fixed with 4% PFA for 10 min, and then washed twice in PBS. Sections were then permeabilized with 0.2% Triton-X-100 in PBS for 5 min and blocked in 3% BSA, 0.2% Triton-X-100 in PBS for 30 min, before incubation in the anti-human TSR rabbit polyclonal antibody (described above) at a 1:200 dilution (in blocking buffer) for 4 hr at 4°C. Slides were subsequently washed three times for 5 min each with 0.2% Triton-X-100 in PBS at rt. Alexa Fluor 546 (Molecular Probes Cat# A-11030 RRID:AB_144695) secondary antibody was diluted in blocking buffer (1:150 dilution) and added to the slides for 1 hr in the dark at rt. Slides were washed three times in the dark for 5 min with 0.2% PBS/Triton-X-100. A drop of Vectashield DAPI hard set mounting medium was then used to stain nuclei and mount cover slips onto slides. Immunofluorescence was visualized using a Zeiss LSM 510 Meta microscope through a Plan-APOCHROMAT 63x/1.4 Oil DIC objective lens. The fluorescence excitation/emission was 556/573 nm for Alexa Fluor 546 and 543/600–700 nm for *Symbiodinium* chlorophyll autofluorescence (see *Supplementary file 2*).

## Experimental manipulation of anemones

In preparation for experimental manipulation, individual anemones were placed in 24-well plates in 2.5 ml FSW and acclimated for 4 days, with FSW replaced daily. During this time, aposymbiotic anemones were maintained in darkness, and symbiotic anemones were maintained in an incubator at 26°C under the light regime described above. Animals were not fed during the experimental period.

Aposymbiotic anemones were experimentally inoculated with *S. minutum* cells and colonization success was determined by quantifying the number of symbionts present in host tissues (see below). Experimental treatments were initiated 2 hr prior to colonization with *S. minutum*. For inoculation, cultured *S. minutum* cells were added to each well to a final concentration of $2 \times 10^5$ cells ml$^{-1}$. After incubation with dinoflagellate cells for 4 hr, anemones were washed twice in FSW and experimental treatments were refreshed. Well-plates were then placed back into an incubator at 26°C under the light regime described above.

### Addition of anti-human TSR rabbit polyclonal antibody during onset of symbiosis

To investigate the effects of blocking TSR domains at the onset of symbiosis, aposymbiotic anemones were incubated with the rabbit anti-human TSR polyclonal antibody as described above. Anemones were incubated for 2 hr prior to inoculation with *S. minutum* in anti-human TSR (Santa Cruz Biotechnology Cat# sc-14013 RRID:AB_2201952), at a concentration of 0.5 µg antibody ml$^{-1}$ FSW. Control animals were given fresh FSW at the same time. For inoculation, cultured *S. minutum* cells were added to each well, to a final concentration of $2 \times 10^5$ cells ml$^{-1}$. After incubation with dinoflagellate cells for 4 hr, anemones were washed twice in FSW and experimental treatments were refreshed. Well-plates were then placed back into an incubator at 26°C under the light regime described above. Anemones were sampled at 48, 72, 96 and 120 hr post-inoculation to measure colonization success. Colonization success was determined by quantifying the number of symbionts present in host tissues (detailed below). Treatment conditions of these animals were refreshed once every 24 hr.

### Addition of human thrombospondin-1 protein

To investigate the effect of TSR proteins on dinoflagellate colonization success, soluble human thrombospondin-1 protein (thrombospondin human platelet, Athens Research and Technology, #:16-20-201319) was added to aposymbiotic anemones at a concentration of 25 µg ml$^{-1}$ FSW. All

other aspects of this experiment were identical to those described for the addition of anti-human TSR.

## Addition of synthetic TSR peptides

To investigate whether native *A. pallida* TSR domains would produce a similar effect to human thrombospondin protein, anemones were incubated in synthetic TSR peptides at a concentration of 150 µg ml$^{-1}$ FSW. Several studies have used TSR peptide fragments to investigate the binding sites of specific receptors such as SRB1 (*Li et al., 1993*; *Tolsma et al., 1993*; *Karagiannis and Popel, 2007*; *Cano et al., 2009*). The putative TSR domain from *A. pallida* contains multiple binding motifs - WXXWXXW, CSVTCG and GVQTRLR - which are all known to bind glycosaminoglycans and class B scavenger receptors in humans. Two separate peptides were designed (*Figure 8B*). Peptide 1 was identical to TSR domain two from the predicted protein *A. pallida* comp25690 (taken from an *A. pallida* transcriptome [*Lehnert et al., 2012*]). For Peptide 2, the cysteine residues in Peptide 1 were substituted with alanine residues to avoid peptide self-adhesion and hence loss of adhesion to target molecules. Peptides were designed according to peptide design guidelines (at www.biomatik.com version 3, RRID:SCR_008944). All other aspects of this experiment were identical to those described for the addition of anti-human TSR.

## Assessing colonization success using confocal microscopy

Colonization success was assessed fluorometrically with a Zeiss LSM 510 Meta confocal microscope, following the methods detailed elsewhere (*Detournay et al., 2012*; *Neubauer et al., 2016*). Colonization success was expressed as the percent of pixels with an autofluorescence intensity above the background intensity. Each experimental treatment had a sample size of three anemones *per* treatment and time-point, with percent colonization taken as a mean of three to four tentacles *per* anemone. Three untreated symbiotic anemones (three to four tentacles *per* anemone) were examined to determine a baseline colonization level for symbiotic anemones. The sample size was limited by both the supply of anemones as well as the number of anemones that could be processed for confocal microscopy at each time point.

## Statistical analysis of colonization success

The statistical significance of colonization success under the treatments described above was assessed using a mixed-effects analysis-of-variance model. As measures on multiple samples (i.e., tentacles) *per* anemone violate independence assumptions, we treated 'anemone' as a random effect to account for correlation among samples within anemones. Main effects included time (in hours) and treatment, and their interaction was estimated to account for differences between treatments at each time point. The full model can be written as:

$$y_{i,j} = \beta X_i + \mu_j + \epsilon_{i,j}$$

Here, $y_{i,j}$ is the logarithm of percent colonization of tentacle *i* within anemone *j*, $\beta$ is a vector of effects to be estimated, *X* is a design matrix encoding the treatment and time point, as well as interaction term contrasts, $\mu_j$ is a normally distributed random effect for anemone *j*, and $\epsilon_{i,j}$ are normally distributed residuals. Contrasts were specified between each treatment and controls at each time-point to assess statistical significance of treatment effects over time, using Tukey's *post-hoc* test to account for multiple comparisons. The model was estimated using the NLME package (*Pinheiro et al., 2016*) for the statistical computing software R (*R-Core-Team, 2012*) (RRID:SCR_001905, www.R-project.org). All datasets and code to reproduce statistical analyses and figures are given as supplementary materials (*Figure 5—source data 1–6*, and Supplementary *Source code 1*).

## qPCR of TSR-domain-containing proteins

To investigate the specific TSR proteins that are involved in the onset of symbiosis, gene expression of two sequences obtained from the bioinformatics searches of the *A. pallida* genome, Ap_Sema5 and Ap_Trypsin-like was measured using quantitative PCR (qPCR). First, to confirm the genome assembly, primers for each sequence were designed using Primer3plus (RRID:SCR_003081, http://primer3plus.com/cgi-bin/dev/primer3plus.cgi) to amplify overlapping 700–900 bp fragments (*Supplementary file 3*). PCR for each primer set was performed using the Go Taq Flexi kit

(Promega, Madison, WI) with the following protocol: 94°C for 3 min, 35 cycles of 94°C for 45 s, annealing temperature for 45 s, and 72°C for 1 min, followed by a final extension at 72°C for 10 min. PCR products were cleaned using the QiaQuick PCR purification kit (Qiagen, Valencia CA) and sequenced on the ABI 3730 capillary sequence machine in the Center for Genome Research and Biocomputing (CGRB) at Oregon State University. Sequences obtained were aligned to the original genome sequence using Geneious v 7.1.8 (RRID:SCR_010519, *Kearse et al., 2012*) to verify amplification of the correct sequence and ensure that overlapping regions between fragments displayed high similarity. If a region varied greatly from the genome, the region was re-sequenced for confirmation before moving forward. Ap_Trypsin-like contained a region that was different than AIPGENE 1852, and therefore this sequence has been submitted to GenBank (accession # KY807678). qPCR primers for products between 100–200 bp with an annealing temperature of 60°C were designed using Primer3 Plus (*Supplementary file 4*), and the products were amplified and sequenced as previously described to confirm the correct amplicon. The efficiency of each primer set was tested to ensure that it was at least 90%.

To investigate the expression of Ap_Sema5 and Ap_Trypsin-like at the onset of symbiosis, qPCR was performed on samples from a previous experiment in which aposymbiotic specimens of *A. pallida* were inoculated with *S. minutum* strain CCMP830 (*Poole et al., 2016*). The two treatment groups used in this study were aposymbiotic animals that were inoculated with symbionts ('inoc') and aposymbiotic animals that received no symbionts and remained aposymbiotic for the duration of the experiment ('apo'). The animals used in this study were sampled at 12, 24, and 72 hr post-inoculation (n = 3 for each time point and treatment combination) Symbiont quantification data indicated symbionts were taken up by 24 hr post-inoculation and levels continued to increase between 24 and 72 hr (*Poole et al., 2016*). The anemones were washed at 24 hr and therefore the increase between 24 and 72 hr can be attributed to symbiont proliferation within the host. Therefore, the time points selected represent a period in which symbionts were actively engaging in recognition and phagocytosis by host cells (12 and 24 hr) and as symbionts were proliferating within the host (72 hr). qPCR plates were run as previously described (*Poole et al., 2016*) using the ABI PRISM 7500 FAST, and resulting Ct values were exported from the machine. Triplicates were averaged and the expression of target genes was normalized to the geometric mean of the reference genes (L10, L12, and PABP). To calculate the $\Delta\Delta$ Ct, the normalized value for each sample was subtracted from the average normalized value of a reference sample, the apo at each time-point. The resulting relative quantities on the $\log_2$ scale were used for statistical analysis using R version 3.2.1 (RRID:SCR_001905, *R Core Team, 2015*). Identical linear models were used to test the hypothesis of no significant difference in gene expression between 'apo' and 'inoc' anemones for both genes. The model was identical to the statistical model described above, but did not include a random effect. A two-way ANOVA was run to test for statistical significance of treatment effects, followed by Tukey's *post hoc* test for pairwise comparisons. All datasets and code to reproduce statistical analyses are given as supplementary materials (*Figure 6—source data 1* and Supplementary *Source code 1*).

## Acknowledgements

We thank Eli Meyer for the reassembly of the *A. pallida* transcriptome, Camille Paxton for immunohistochemistry advice and Anne LaFlamme for providing parasitology perspectives. We wish to acknowledge the Confocal Microscopy Facility at the Center for Genome Research and Biocomputing at Oregon State University. This work was partially supported by a grant from the National Science Foundation to VMW (IOB0919073). EFN was supported by a Commonwealth Doctoral Scholarship and a Faculty of Science Strategic Research Grant from Victoria University of Wellington. KT was supported on SURE Science summer fellowship from the OSU College of Science.

## Additional information

### Funding

| Funder | Grant reference number | Author |
|---|---|---|
| National Science Foundation | IOB0919073 | Virginia M Weis |

| Victoria University of Wellington | Emilie-Fleur Neubauer |
|---|---|
| Oregon State University | Kenneth Tan |

The funders had no role in study design, data collection and interpretation, or the decision to submit the work for publication.

## Author contributions

E-FN, Conceptualization, Formal analysis, Funding acquisition, Investigation, Methodology, Writing—original draft, Writing—review and editing; AZP, Investigation, Methodology, Writing—review and editing; PN, Data curation, Formal analysis, Methodology; OD, Investigation, Methodology; KT, Formal analysis, Investigation; SKD, Supervision, Funding acquisition, Methodology, Project administration, Writing—review and editing; VMW, Conceptualization, Supervision, Funding acquisition, Methodology, Writing—original draft, Project administration, Writing—review and editing

## Author ORCIDs

Philipp Neubauer, http://orcid.org/0000-0002-4150-848X
Virginia M Weis, http://orcid.org/0000-0002-1826-2848

# Additional files

## Supplementary files

• Supplementary file 1. Tabulated TSR sequences identified from searches of six cnidarian and two dinoflagellate resources and TSR sequences from other organisms used in this study. Sequences are sorted by protein type or source organism.

• Supplementary file 2. Summary of fluorescent dyes and their excitation and emission wavelengths used for confocal microscopy

• Supplementary file 3. Primers for initial PCR of TSR sequences.

• Supplementary file 4. Primers used for qPCR of Ap_Sema5 and Ap_Trypsin-like amplicons.

• Source code 1. R-code for statistical analyses performed for data displayed in *Figures 5* and *6*.

## Major datasets

The following previously published datasets were used:

| Author(s) | Year | Dataset title | Dataset URL | Database, license, and accessibility information |
|---|---|---|---|---|
| Putnam NH, Srivastava M, Hellsten U, Dirks B, Chapman J, Salamov A, Terry A, Shapiro H, Lindquist E, Kapitonov VV, Jurka J, Genikhovich G, Grigoriev IV, Lucas SM, Steele RE, Finnerty JR, Technau U, Martindale MQ, Rokhsar DS | 2007 | Sea Anemone Genome Reveals Ancestral Eumetazoan Gene Repertoire and Genomic Organization | https://www.ncbi.nlm.nih.gov/bioproject/PRJNA19965 | Publicly available at NCBI BioProject (accession no. PRJNA19965). This work used the following resource built from this data: http://genome.jgi.doe.gov/Nemve1/Nemve1.home.html |
| Putnam NH, Srivastava M, Hellsten U, Dirks B, Chapman J, Salamov A, Terry A, Shapiro H, Lindquist E, Kapi- | 2007 | Sea anemone genome reveals ancestral eumetazoan gene repertoire and genomic organization | https://www.ncbi.nlm.nih.gov/bioproject/PRJNA12581 | Publicly available at NCBI BioProject (accession no. PRJNA12581). This work used the following resource |

| | | | | |
|---|---|---|---|---|
| tonov VV, Jurka J, Genikhovich G, Grigoriev IV, Lucas SM, Steele RE, Finnerty JR, Technau U, Martindale MQ, Rokhsar DS | | | | built from this data: http://genome.jgi.doe.gov/Nemve1/Nemve1.home.html |
| Kitchen SA, Crowder CM, Poole AZ, Weis VM, Meyer E | 2015 | Data from: De novo assembly and characterization of four anthozoan (phylum Cnidaria) transcriptomes | http://dx.doi.org/10.5061/dryad.3f08f | Available at Dryad Digital Repository under a CC0 Public Domain Dedication. This work used the following resource built from this data: http://people.oregonstate.edu/~meyere/data.htm |
| Lehnert EM, Burriesci MS, Pringle JR | 2012 | Developing the anemone Aiptasia as a tractable model forcnidarian-dinoflagellate symbiosis: the transcriptome of aposymbiotic A. pallida | https://www.ncbi.nlm.nih.gov/sra/SRR696721 | Publicly available at the NCBI Sequence Read Archive (accession no. SRR696721). This work used the following resource built from this data: http://pringlelab.stanford.edu/projects.html |
| Baumgarten S, Simakov O, Esherick LY, Liew YJ, Lehnert EM, Michell CT, Li Y, Hambleton EA, Guse A, Oates ME, Gough J, Weis VM, Aranda M, Pringle JR, Voolstra CR | 2015 | The genome of Aiptasia, a sea anemone model for coral symbiosis | https://www.ncbi.nlm.nih.gov/bioproject/PRJNA261862 | Publicly available at NCBI BioProject (accession no. PRJNA261862). This work uses the following resource built from this data: http://aiptasia.reefgenomics.org/ |
| Shinzato C, Shoguchi E, Kawashima T, Hamada M, Hisata K, Tanaka M, Fujie M, Fujiwara M, Koyanagi R, Ikuta T, Fujiyama A, Miller DJ, Satoh N | 2011 | Using the Acropora digitifera genome to understand coral responses toenvironmental change | https://www.ncbi.nlm.nih.gov/bioproject/PRJNA314803 | Publicly available at NCBI BioProject (accession no. PRJNA314803). This work uses the following resource built from this data: http://marinegenomics.oist.jp/coral/viewer/info?project_id=3 |
| Shinzato C, Shoguchi E, Kawashima T, Hamada M, Hisata K, Tanaka M, Fujie M, Fujiwara M, Koyanagi R, Ikuta T, Fujiyama A, Miller DJ, Satoh N | 2011 | Using the Acropora digitifera genome to understand coral responses to environmental change | https://www.ncbi.nlm.nih.gov/bioproject/PRJDA67425 | Publicly available at NCBI BioProject (accession no. PRJDA67425). This work uses the following resource built from this data: http://marinegenomics.oist.jp/coral/viewer/info?project_id=3 |
| Moya A, Huisman L, Ball EE, Hayward DC, Grasso LC, Chua CM, Woo HN, Gattuso J-P, Forêt S, Miller DJ | 2012 | Whole transcriptome analysis of the coral Acropora millepora reveals complex responses to $CO_2$-driven acidification during the initiation of calcification | https://www.ncbi.nlm.nih.gov/bioproject/PRJNA74409 | Publicly available at NCBI BioProject (accession no. PRJNA74409). This work uses the following resource built from this data: http://www.bio.utexas.edu/research/matz_lab/matzlab/Data.html |

| | | | | |
|---|---|---|---|---|
| Shoguchi E, Shinzato C, Kawashima T, Gyoja F, Mungpakdee S, Koyanagi R, Takeuchi T, Hisata K, Tanaka M, Fujiwara M, Hamada M, Seidi A, Fujie M, Usami T, Goto H, Yamasaki S, Arakaki N, Suzuki Y, Sugano S, Toyoda A, Kuroki Y, Fujiyama A, Medina M, Coffroth MA, Bhattacharya D, Satoh N | 2013 | Draft assembly of the Symbiodinium minutum nuclear genome reveals dinoflagellate gene structure | https://www.ncbi.nlm.nih.gov/bioproject/PRJDB732 | Publicly available at NCBI BioProject (accession no. PRJDB732). This work uses the following resource built from this data: http://marinegenomics.oist.jp/symb/viewer/info?project_id=21 |
| Aranda M, Li Y, Liew YJ, Baumgarten S, Simakov O, Wilson MC, Piel J, Ashoor H, Bougouffa S, Bajic VB, Ryu T, Ravasi T, Bayer T, Micklem G, Kim H, Bhak J, LaJeunesse TC, Voolstra CR | 2016 | Genomes of coral dinoflagellate symbionts highlight evolutionary adaptations conducive to a symbiotic lifestyle | https://www.ncbi.nlm.nih.gov/bioproject/PRJNA292355 | Publicly available at NCBI BioProject (accession no. PRJNA292355). This work uses the following resource built from this data: http://smic.reefgenomics.org/ |

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
