## [Decision Letter]

Thank you for submitting your article "A diverse host TSR protein repertoire promotes symbiont colonization during onset of cnidarian-dinoflagellate symbiosis" for consideration by *eLife*. Your article has been reviewed by two peer reviewers, and the evaluation has been overseen by a Reviewing Editor and Detlef Weigel as the Senior Editor. The reviewers have opted to remain anonymous.

The Reviewing Editor has drafted this decision to help you prepare a revised submission.

Summary:

This is an excellent, very well documented contribution to understanding the molecular mechanism of the acquisition and retention of zooxanthellae in cnidarian hosts.

Essential revisions:

Included below are the full reviews. Please make the suggested revisions to accommodate the reviewers' concerns.

*Reviewer #2:*

I enjoyed reading this manuscript, it brings in new and exciting evidence for the process of symbiont acquisition in cnidarians. The evidence is overwhelming – I was especially intrigued with the "Blocking TSR domains inhibits symbiont uptake by host anemones" section where anemones pre-incubated in an antihuman thrombospondin reviled an almost complete arrest of colonization. In addition, the "super colonization" that was induced via addition of synthetic TSR peptides and exogenous human thrombospondin-1 was also a very strong find supporting the authors hypothesis. The authors include another crucial piece of evidence as they followed gene expression of Ap_Sema5 and Ap_Trypsin-like at from the commencement of the symbiosis for 72 hours, increased expression and then a decline following the uptake of the symbionts and the aftermath – represented as a decline in expression.

There are multiple important finds in this manuscript and they are supported by the evidence:

Diversity of anthozoan TSR proteins, that led to the interesting view by the authors that it may be related to the complex management of their microbiomes and not solely the *Symbiodinium* part of the symbiosis. The authors hypothesis that the TSR-only repertoire similarity to the vertebrate complement protein properdin places it as a prime candidate been a specific pattern recognition receptor (PRR) for *Symbiodinium* (the key was a decrease in expression in the later stages of symbiosis vs an aposymbiotic larvae.

In short I didn't find any major or even minor follows in this work and as wrote I did enjoy this manuscript very much.

*Reviewer #3:*

Neubaeur and coauthors provide an elegant study of the diversity and functional role of thrombospondin-type-1 repeat (TSR) domain proteins in the establishment of the coral-dinoflagellate symbiosis. They hypothesize that TSR proteins are involved in symbiont uptake through the promotion of phagocytosis. More specifically, they may act as pattern recognition receptors, and/or activate and stabilize a proteolytic complex that attaches to the surface of cells and marks them for phagocytosis. The study takes a rigorous and multi-pronged approach including: bioinformatics analysis for identification and description of the TSR protein repertoire in six anthozoans and *Symbiodinium* minutum, identification of TSR proteins through immunoblots and confocal microscopy visualization of immunofluorescence, functional analysis to track inoculation while blocking TSR domains and enhancing TSR protein abundance through exogenous delivery, and QPCR of specific TSR proteins post inoculation.

This study delivers substantial advancement in a very critical gap in our knowledge of the basic biology of symbiotic cnidarians. The work provides a systematic analysis that identifies a promising set of proteins with regards to their functional role in symbiotic initiation and wealth of testable hypotheses for further essential functional work symbioses. Given the use of human to apicomplexan endoparasite comparisons and references, and implication for symbioses in ecosystem engineering corals, this paper should have a wide reach while transforming and re-invigorating the field of cnidarian symbiosis and furthering the potential for functional genomic approaches.

As written there is a lack of rationale for the six anthozoan species chosen and single dino, given other genomic resources are available. All of the symbiotic species selected here are horizontal transmitters, while genomic data are available for investigation of the TSR protein repertoire in at least three vertical transmitters (with different symbiosis initiation dynamics) and another two dinos (which have been shown to have very strong genomic divergence).

Porites lutea PRJEB6884 https://www.ncbi.nlm.nih.gov/biosample?LinkName=bioproject_biosample_all&from_uid=327630

Stylophora pistillata and Seriatopora sp.https://elifesciences.org/content/5/e13288

Symbiodinium microadriaticum = http://smic.reefgenomics.org/

Symbiodinium kawagutii = Lin, S. et al. Science 350, 691-694

The manuscript structuring has resulted in a disjointed rationale with respect to the gene expression work and gene choice. I would recommend a reorganization of some of the gene selection rationale in the Methods and Discussion to earlier in the manuscript. Additionally, the rationale for investigation of and conclusions for the findings on Ap_Trypsin-like are lacking throughout.

The temporal frequency of sampling for gene expression during the symbiotic initiation experiment are fairly coarse, without strong justification for time points selected with regards to documented timing of ingestion and phagocytosis. I suggest consideration of further temporal investigation of these genes of interest could be easily assessed on existing transcriptomics data thereby providing additional time points (e.g., Mohamed et al., 2016) and a better understanding of temporal dynamics.

Regarding the functional experiments and host colonization, how is colonization through phagocytosis versus cell division differentiated here? What are natural cell division rates relative to the increased densities observed in the experiments? Is it clear that the increase in cell densities was through primary promotion of phagocytosis?

---

## [Author Response]

*[…] Reviewer #3: […] As written there is a lack of rationale for the six anthozoan species chosen and single dino, given other genomic resources are available. All of the symbiotic species selected here are horizontal transmitters, while genomic data are available for investigation of the TSR protein repertoire in at least three vertical transmitters (with different symbiosis initiation dynamics) and another two dinos (which have been shown to have very strong genomic divergence).*

*Porites lutea PRJEB6884 https://www.ncbi.nlm.nih.gov/biosample?LinkName=bioproject_biosample_all&from_uid=327630
*

*Stylophora pistillata and Seriatopora sp.https://elifesciences.org/content/5/e13288
*

*Symbiodinium microadriaticum = http://smic.reefgenomics.org/
*

*Symbiodinium kawagutii = Lin, S. et al. Science 350, 691-694*

We have provided rationale for the species used in this study (subsection “Colonization experiments implicate the TSR domain in symbiosis establishment”, fifth paragraph). In addition, we have expanded our searches to include two additional species that were suggested including genomes of the vertical transmitter *Stylophora pistillata* and *Symbiodium microadriaticum* (see Figure 2 and Figure 3 and the [Supplementary-material SD8-data]).

*The manuscript structuring has resulted in a disjointed rationale with respect to the gene expression work and gene choice. I would recommend a reorganization of some of the gene selection rationale in the Methods and Discussion to earlier in the manuscript. Additionally, the rationale for investigation of and conclusions for the findings on Ap_Trypsin-like are lacking throughout.*

We have moved the rationale for the gene selection to the beginning of the qPCR Results section (Subsection “Blocking TSR domains inhibits host colonization by symbiont”). Additionally, we have expanded upon our rationale for the selection of Ap_Trypsin and the explanation of the results for this gene in the conclusion section (Discussion).

*The temporal frequency of sampling for gene expression during the symbiotic initiation experiment are fairly coarse, without strong justification for time points selected with regards to documented timing of ingestion and phagocytosis. I suggest consideration of further temporal investigation of these genes of interest could be easily assessed on existing transcriptomics data thereby providing additional time points (e.g., Mohamed et al., 2016) and a better understanding of temporal dynamics.*

We have added information regarding time point selection within the qPCR Methods section (subsection “Statistical analysis of colonization success”). To determine whether Ap_Sema5 and Ap_Trypsin-like are differentially expressed in onset of symbiosis transcriptomic data sets, we examined a recent study on *A. pallida* larvae (Wolfowicz et al., 2016) and the suggested *Acropora digitifera* study (Mohamed et al., 2016). For *A. pallida*, neither Ap_Sema5 or Ap_Trypsin-like was in the supplemental list of differentially expressed genes (at 5-6 days post-inoculation). However, several TSR protein genes were downregulated in symbiotic larvae, along with a non-TSR containing semaphorin (Semaphorin-3E). We have therefore included this information in the manuscript as it supports the trends we observed in our qPCR data (Subsection “Colonization experiments implicate the TSR domain in symbiosis establishment”). For the *A. digitifera* study, no overall differences in gene expression observed at 12 and 48 h post-inoculation. At 4 hours post- inoculation, there was a trend for both genes of greater counts in the symbiotic samples vs the non-symbiotic larvae. However, our understanding is that these are raw count data and have not been normalized to account for read number variation across transcriptomes. Therefore, we could not reach a definitive conclusion regarding their expression levels without substantial further analysis. We agree that including more time points, especially prior to 12 h post-inoculation would help us to understand the role of these genes in symbiosis and therefore represents an interesting extension of this project for future exploration.

*Regarding the functional experiments and host colonization, how is colonization through phagocytosis versus cell division differentiated here? What are natural cell division rates relative to the increased densities observed in the experiments? Is it clear that the increase in cell densities was through primary promotion of phagocytosis?*

In this study, colonization through phagocytosis versus cell division is not differentiated. Essentially we are interested in overall symbiont levels within the host, regardless of the mechanisms they used to enter host cells. Therefore, we have attempted to frame our discussion in terms of overall host colonization levels and have replaced the term uptake with colonization in the text where appropriate. We have preliminary evidence that the addition of human Thrombospondin increases symbiont cell division rates within the host and these data would suggest that the increase in symbiont numbers is related to increased proliferation within the host, but does not rule out the possibility that it also influences the rate of phagocytosis. However, these data are from a pilot study and have not yet been replicated and therefore we do not feel comfortable including it in the study. Future studies that specifically quantify mitotic index and phagocytosis rates in response to TSR proteins would better be able to differentiate these two processes, but we feel this is beyond the scope of our study.